# RAM: Retrieval-Based Affordance Transfer for Generalizable Zero-Shot Robotic Manipulation

**Yuxuan Kuang**[1,2*], **Junjie Ye**[1*], **Haoran Geng**[2,3*], **Jiageng Mao**[1], **Congyue Deng**[3],
**Leonidas Guibas**[3], **He Wang**[2], **Yue Wang**[1]

[1]University of Southern California, [2]Peking University, [3]Stanford University
[*]Equal contributions

**Abstract:** This work proposes a retrieve-and-transfer framework for zero-shot robotic manipulation, dubbed RAM, featuring generalizability across various objects, environments, and embodiments. Unlike existing approaches that learn manipulation from expensive in-domain demonstrations, RAM capitalizes on a retrieval-based affordance transfer paradigm to acquire versatile manipulation capabilities from abundant out-of-domain data. First, RAM extracts unified affordance at scale from diverse sources of demonstrations including robotic data, human-object interaction (HOI) data, and custom data to construct a comprehensive affordance memory. Then given a language instruction, RAM hierarchically retrieves the most similar demonstration from the affordance memory and transfers such out-of-domain 2D affordance to in-domain 3D executable affordance in a zero-shot and embodiment-agnostic manner. Extensive simulation and real-world evaluations demonstrate that our RAM consistently outperforms existing works in diverse daily tasks. Additionally, RAM shows significant potential for downstream applications such as automatic and efficient data collection, one-shot visual imitation, and LLM/VLM-integrated long-horizon manipulation. For more details, please check our website at `https://yuxuank.com/RAM/`.

**Keywords:** Hierarchical Retrieval, Affordance Transfer, Zero-Shot Robotic Manipulation, Visual Foundation Models

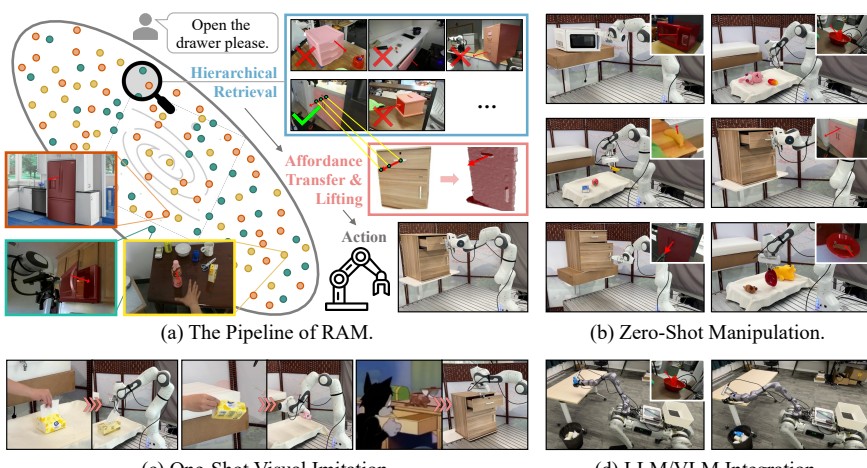

(a) The Pipeline of RAM.

(b) Zero-Shot Manipulation.

(c) One-Shot Visual Imitation.

(d) LLM/VLM Integration.

Figure 1: (a) We extract unified affordance representation from in-the-wild multi-source demonstrations, including robotic data, HOI data, and custom data, to construct a large-scale affordance memory. Given language instructions, RAM hierarchically retrieves and transfers the 2D affordance from memory and lifts it to 3D for robotic manipulation. (b-d) Our framework shows robust generalizability across diverse objects and embodiments in various settings.

8th Conference on Robot Learning (CoRL 2024), Munich, Germany.

# 1 Introduction

A longstanding goal in robot learning is to develop a generalist robot agent capable of performing diverse robotic manipulation tasks on common household objects in real-world settings. Crucially, such an agent must learn to generalize to manipulate *unseen* objects in *unseen* environments with *unseen* embodiments. Traditional approaches to achieving this goal often involve manually collecting extensive expert demonstrations through teleoperation, scripted policies, and similar methods, followed by imitation learning from these datasets [1, 2]. However, these methods are costly and labor-intensive, requiring significant human effort and prior knowledge of various objects and tasks. As a result, the scarcity of usable real-world data challenges the generalization of policies trained on these datasets to unseen objects or environments.

On the other hand, apart from real-world and synthetic robotic datasets [1, 2, 3, 4, 5, 6, 7, 8] that agents can directly learn manipulation policies from, there exist vast amounts of out-of-domain data rich with actionable knowledge, ranging from Hand-Object Interaction (HOI) data [9, 10, 11, 12], to Internet-scale videos of daily activities, AI-generated videos [13], and even sketches [14]. Despite efforts to leverage these data sources for robotic manipulation [15, 16, 17], learning from out-of-domain samples remains elusive due to significant domain shifts. Addressing this issue requires rethinking how to unify and utilize the actionable knowledge hidden within these heterogeneous and often noisy data sources. To that end, we identify that the key is to represent the actionable knowledge as **transferrable affordance**, *i.e.*, 'where' and 'how' to act [18, 19]. By efficiently extracting and transferring these affordances from diverse data sources to our target domain, we can overcome domain-specific hurdles and data scarcity. In addition, the fast development of generalizable visual foundation models (VFMs), trained on enormous Internet images, offers a streamlined solution to seamlessly connecting diverse data realms with our target domain, facilitating the distillation of rich affordance from these data sources to enable generalizable zero-shot robotic manipulation.

Therefore, we introduce **RAM**, a **R**etrieval-based **A**ffordance transfer approach for generalizable zero-shot robotic **M**anipulation. RAM utilizes a retrieve-and-transfer paradigm for everyday robotic tasks in a **zero-shot** manner. First, we extract 2D affordance from diverse data sources like robotic datasets, HOI datasets, Internet images, *etc.*, to construct a comprehensive affordance memory. Upon receiving a monocular RGBD observation of an unseen object in an unseen environment, RAM employs an effective hierarchical retrieval pipeline, which selects the most similar demonstration from the affordance memory and leverages VFMs to transfer the pixel-aligned 2D affordance to the target unseen domain. To convert the 2D affordance to executable robotic actions, we further develop a sampling-based affordance lifting module. This module lifts the 2D affordance to a 3D representation that includes a contact point and a post-contact direction, which is directly executable by various robotic systems using off-the-shelf grasp generators [20, 21] and motion planners [22, 23].

Extensive experiments in both simulation and the real world show that our method surpasses existing works by a large margin (§4.3), thereby confirming its effectiveness and superiority in leveraging large out-of-domain data. Notably, our method is embodiment-agnostic and data-efficient—qualities that prior methods are struggling with—which is substantiated by abundant experiments conducted across various robotic platforms. Furthermore, we showcase the versatility of our retrieval-based framework through its applications in several key areas in §4.5, such as automatic and efficient data collection, one-shot visual imitation conditioned on human preference, and seamless integration with LLMs/VLMs for long-horizon tasks with free-form human instructions. In summary, the key contributions of our proposed RAM are three-fold:

- We propose a retrieval-based affordance transfer framework for zero-shot robotic manipulation, significantly outperforming prior works, both in simulation and the real world. Our key insight is to transfer affordances in 2D, followed by a module to lift them into 3D for direct execution.
- We propose a scalable module for extracting unified affordance information from diverse out-of-domain heterogeneous data for retrieval. This module can potentially generalize to other robotic tasks beyond the manipulation considered by this work.
- Our pipeline enables a variety of intriguing downstream applications, including policy distillation, one-shot visual imitation, and LLM/VLM integration to facilitate future research.

## 2 Related Works

**Affordance for Robotic Manipulation.** Visual affordance, which indicates where and how to interact with diverse objects from visual inputs, plays an important role in robotic manipulation thanks to its simplicity and interpretability. Point cloud-based methods [18, 19, 24, 25] focus on learning a point-wise score map as the representation of affordance and utilizing it to regress end-effector poses. However, these methods face severe sim-to-real gaps due to the depth noises of RGBD cameras in the real world. Other series of works such as HOI-Forecast [15] and VRB [16] aim to learn affordance from human videos in an end-to-end manner. However, the scarcity of training data diversity hinders their generalization to unseen tasks and objects. Compared to these existing works, our method leverages the unified affordance representation extracted from diverse data sources and is thus generalizable to a wide range of unseen objects and environments in the real world.

**Zero-Shot Robotic Manipulation.** Due to the issues mentioned above, how to manipulate diverse kinds of objects in a zero-shot manner remains a challenging topic. Existing works [26, 27, 28, 29, 30, 31, 32, 33, 34, 35] attempt to solve this problem by leveraging the reasoning abilities of LLMs/VLMs. However, they heavily rely on pre-programmed heuristic action primitives to execute low-level tasks like "grasping a handle". Other works [36, 37, 38] leverage intermediate representations as conditions to the policy to achieve data-efficient imitation learning and can generalize to in-domain tasks without direct demonstrations. However, these methods still require collecting in-domain demonstration for test-time training and thus cannot be generalized to in-the-wild domains. Compared to these works, our method is training-free, neglecting the need for heuristic policies or test-time training.

**Learning from Demonstrations.** Recent years have witnessed the rapid progress of imitation learning (IL). Multiple teleoperation systems and algorithms [39, 40, 41, 42] have shown great capabilities to imitate human demonstrations smoothly. There are also methods [43, 1, 2, 44] that leverage VLMs or co-train data priors to reduce the requirement for demonstrations. Other series of works [45, 46, 47, 48, 49, 50, 51, 52, 53, 54] learn manipulation from demonstrations based on retrieving in-domain demonstrations for IL or direct trajectory replay. However, these methods fail to generalize to the wild without in-domain demonstrations. Recent work Robo-ABC [55] tries to leverage CLIP [56] and Stable Diffusion [57] to retrieve and transfer contact points, but it is limited to HOI and table-top grasping scenarios and cannot perform object manipulation in the 3D space. Our method, instead, not only makes the most of existing diverse out-of-distribution data sources to guide affordance transfer in 3D but also generalizes to a diverse set of in-the-wild tasks in a zero-shot manner.

## 3 Method

To address the in-domain data scarcity, the core of RAM involves transferring out-of-domain knowledge to our unseen target domain. To achieve so, we introduce a retrieve-and-transfer approach that capitalizes on visual affordance. In contrast to existing works [16, 55] that only predicts 2D affordance in the pixel space, RAM generates executable 3D affordance $\mathcal{A}^{3D} = (c^{3D}, \tau^{3D})$ where $c^{3D}, \tau^{3D} \in \mathbb{R}^3$ are the 3D contact point and the post-contact direction respectively.

First, RAM extracts affordance information from diverse out-of-domain data sources, constructing an affordance memory $\mathcal{M}$ (§3.1) in source domain $\mathcal{S}$. Then, given a monocular RGBD image $(I^{\mathcal{T}}, D)$ from the novel target domain $\mathcal{T}$ along with a language instruction $L$, RAM leverages a three-step hierarchical retrieval procedure (§3.2) to identify the most similar demonstration from $\mathcal{M}$. Leveraging VFMs, we establish image-pair correspondences to transfer the 2D affordance from the demonstration in source domain $\mathcal{S}$ to our target domain $\mathcal{T}$ (§3.3). Finally, through a sampling-based method, the 2D affordance in the target domain is lifted back to 3D, resulting in the 3D affordance $\mathcal{A}^{3D}$ (§3.4), which can be directly executed through grasp generators [20, 21] and motion planners [22, 23].

### 3.1 Affordance Memory

To achieve generalization across objects, environments, and embodiments, RAM opts for a unified affordance representation that can be easily acquired from diverse data sources in a scalable manner,

constructing a comprehensive affordance memory, $\mathcal{M}$, that encompasses a wide range of skills and objects. This memory integrates subsets of demonstrations from real-world or synthetic robotic data $\mathcal{M}_R$, HOI data $\mathcal{M}_H$, and custom data $\mathcal{M}_C$. To unify affordance information from these varied sources, each entry in the affordance memory includes an object-centric RGB image $I^{\mathcal{S}}$ which is the initial frame of interaction, a set of 2D waypoints $C = (c_0^{2D}, c_1^{2D}, c_2^{2D}, \cdots)$ indicating the contact point $c_0^{2D}$ and post-contact trajectories, along with a task category $T$ expressed in natural language. Therefore, the constructed affordance memory can be denoted as:

$$\mathcal{M} = \mathcal{M}_R \cup \mathcal{M}_H \cup \mathcal{M}_C = \{(I^{\mathcal{S}}, T, C) \mid C = (c_0^{2D}, c_1^{2D}, c_2^{2D}, \cdots)\}. \tag{1}$$

For different data sources, we employ different strategies to extract and annotate affordance:

**Robotic Data.** In the real world or simulators, obtaining camera parameters and robot proprioception is straightforward. This allows us to seamlessly project the end-effector's 3D position onto a 2D image plane. For affordance extraction, we start with the first frame of a rollout for the image $I^{\mathcal{S}}$, ensuring the object is unobscured. Then, we identify the frame when the robot's gripper closes, and use the end-effector's 2D position at this moment as the contact point $c_0^{2D}$. Subsequent frames are used to trace post-contact trajectories. In our experiments, we utilize DROID [2] dataset for robotic data subset construction.

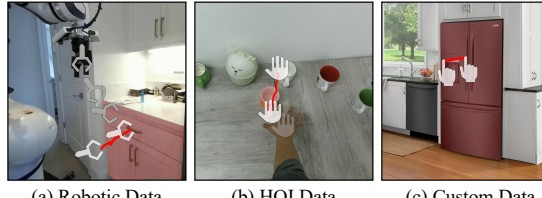

(a) Robotic Data  (b) HOI Data  (c) Custom Data

Figure 2: Affordance annotation of demonstrations from various data sources. We extract unified affordance information automatically (for robotic and HOI data) or with minimal human effort (for custom data).

**HOI Data.** Human demonstrations, compared to robotic data, are often considered more intuitive and efficient. For affordance extraction, we begin by identifying and grounding video segments where relevant events occur. Similar to the approach with robotic data, we select the initial frame from each segment to serve as $I^{\mathcal{S}}$. Then, the contact point and post-contact trajectories are determined by averaging the hand keypoints across frames. We found this simple strategy can generate reliable hand waypoints. We utilize the annotations of video segments and hand keypoints from the HOI4D [12] dataset to construct the subset while those annotations can also be easily acquired through action segmentation [58] and hand-object segmentation [59] techniques.

**Custom Data.** Beyond the automatic collection of affordance information from robotic and HOI data, we also consider custom data that are manually annotated with minimal human effort. This allows us to tailor our manipulation strategies for novel objects by augmenting the affordance memory with newly annotated demonstrations. The annotation process can be realized by selecting the start and end points on an RGB image, followed by automatic interpolation between the two points.

Notably, our affordance memory is built scalably and semi-automatically, featuring manipulation customization with minimal human intervention. Putting all subsets from different data sources together, we easily establish an affordance memory $\mathcal{M}$ spanning diverse objects, environments, and embodiments, enabling our method's generalizability across various settings.

### 3.2 Hierarchical Retrieval

To correctly manipulate a new object, humans often turn to their memory for similar scenarios for guidance. Inspired by how humans think and Retrieval-Augmented Generation (RAG) in LLM inference [60], we proposed a three-step hierarchical retrieval pipeline to effectively retrieve the most similar demonstration from the agent's affordance memory $\mathcal{M}$ in a coarse-to-fine manner, and leverage the demonstration as a hint to guide the manipulation.

**Task Retrieval.** Based on the language instruction $L$ (*e.g.*, *"Please open the drawer to find some utensils."*), we leverage text encoders (*e.g.*, CLIP [56]) or language models (*e.g.*, GPT-4 [61]) to retrieve the task $T$ that the instruction falls within (*"open the drawer"*) and extracts the object name $N$ (*"drawer"*). If the memory doesn't contain the queried object, the model will reason the object geometry and propose a task (or multiple tasks) that potentially shares similar affordance.

**Semantic Filtering.** Then in the retrieved tasks, we can get the source RGB images $\{I^{\mathcal{S}}\}$ of all demonstrations. For each source image, we can calculate its joint similarity with the target observation image $I^{\mathcal{T}}$ and the object name $N$, in the form of

$$\text{similarity} = \cos(\text{CLIP}_\text{v}(I^{\mathcal{S}}), \text{CLIP}_\text{v}(I^{\mathcal{T}})) \cdot \cos(\text{CLIP}_\text{v}(I^{\mathcal{S}}), \text{CLIP}_\text{t}(N)), \qquad (2)$$

where $\text{CLIP}_\text{v}(\cdot)$ and $\text{CLIP}_\text{t}(\cdot)$ are CLIP [56] visual and text encoders, and $\cos(\cdot)$ measures the cosine similarity between two embeddings. Then, we set a threshold to filter out demonstrations that have very low similarities—which can be caused by a lack of intended objects, low-quality segmentation masks, low-light environments, and so on—to improve the robustness of subsequent geometrical retrieval and affordance transfer.

**Geometrical Retrieval.** Large amounts of previous research work [62, 63, 64] reveal the emergent correspondence from large-scale unsupervised visual foundation models [65, 56, 57, 66]. The core concept is that the deep dense feature maps produced by visual foundation models contain rich geometrical and semantic information that can be used for dense keypoint correspondence matching. However, as illustrated in [67, 55], these visual foundation models often struggle to understand the **orientation** of instances. Therefore, it is essential for our method to find a demonstration where the object is oriented similarly to the target image. To this end, after task retrieval and semantic filtering, we calculate Instance Matching Distance (IMD) [67] using Stable Diffusion feature maps to per-

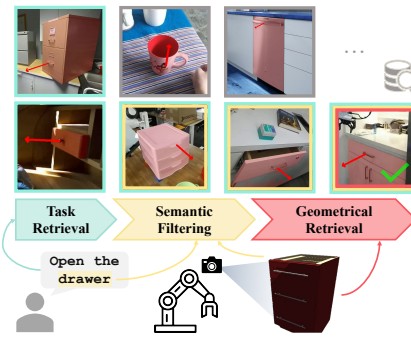

Figure 3: Illustration of our hierarchical retrieval pipeline.

form geometrical retrieval from the remaining demonstrations to find the one with **the most similar viewpoint**. More details can be found in the appendix. By leveraging the hierarchical retrieval, we can retrieve the most suitable demonstration for manipulation, both semantically and geometrically. Experiment results show that it performs better than a single-stage retrieval pipeline, found in Table 3.

### 3.3 2D Affordance Transfer

After getting the most similar demonstration from the affordance memory, we aim to transfer the 2D affordance from the source domain $\mathcal{S}$ to the target domain $\mathcal{T}$ in a generalizable way. Given the source image $I^{\mathcal{S}}$ and its contact waypoints $C$, leveraging dense feature maps by visual foundation models, we first perform per-point correspondence matching to obtain the corresponding waypoints in the target image. We then employ RANSAC algorithm [68] to remove outliers of target waypoints and fit a line in the 2D space. In this way, we obtain the 2D contact point $c^{\text{2D}}$, which is the first transferred waypoint, and the post-contact direction $\tau^{\text{2D}}$, leading to the 2D affordance $\mathcal{A}^{\text{2D}} = (c^{\text{2D}}, \tau^{\text{2D}})$. By doing so, we can establish a robust affordance transfer from $\mathcal{S}$ to $\mathcal{T}$.

### 3.4 Sampling-Based Affordance Lifting

The 2D affordance $\mathcal{A}^{\text{2D}} = (c^{\text{2D}}, \tau^{\text{2D}})$ obtained from the affordance transfer step cannot be directly used in the 3D space. Therefore, we propose a simple yet effective sampling-based method to lift the affordance to 3D, which can be executed by the end-effector in the 3D space.

Based on the contact point $c^{\text{2D}}$ and the partial point cloud generated from the depth map $D$, we back-project the 2D contact point to 3D to get the 3D contact point $c^{\text{3D}}$ using depth $D$. Then we crop the point cloud around $c^{\text{3D}}$ to acquire the local geometry of the contact area. Based on the cropped point cloud, we first estimate the normal vector of each point and then perform K-Means algorithm [69] to cluster these normals and get Top-$K$ cluster centers. Then

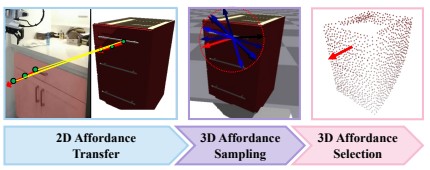

Figure 4: Affordance transfer and lifting.

we back-project these cluster center normal vectors into the 2D space through camera parameters and

| Object | 🗄 | | 📠 | | 🖥 | | 🗄 | 🗄 | 🗑 | 🗑 | 🗋 | ⊻ | 🔋 | AVG |
|---|---|---|---|---|---|---|---|---|---|---|---|---|---|---|
| Task | O | C | O | C | O | C | O | O | O | P | P | P | P | / |
| Where2Act [18] | 2 | 34 | 2 | 54 | 2 | **68** | 2 | 0 | / | / | / | / | / | 20.50 |
| VRB* [16] | 8 | 62 | 6 | 56 | 16 | 66 | 4 | 12 | 10 | 18 | 28 | 44 | 60 | 30.77 |
| Robo-ABC* [55] | 20 | 58 | 22 | 60 | 30 | 46 | 30 | 28 | 26 | 40 | 54 | 66 | 60 | 41.54 |
| **RAM (Ours)** | **38** | **68** | **32** | **76** | **32** | 50 | **66** | **54** | **38** | **46** | **56** | **72** | **64** | **52.62** |

Table 1: Success rates of object manipulation for different methods in the simulation environment. O, C, and P stand for Open, Close, and Pickup, respectively. * denotes necessary modification for 3D adaptation. RAM consistently and significantly outperforms baseline methods in most tasks.

select the one with the least included angle with the 2D post-contact direction $\tau^{2D}$. In this way, we can obtain the 3D affordance $\mathcal{A}^{3D} = (c^{3D}, \tau^{3D})$ for robotic manipulation.

To map the 3D affordance to concrete robot actions, we sample dense grasp proposals on the cropped point cloud leveraging off-the-shelf dense grasp generators [20, 21] and select the closest grasp from $c^{3D}$. After grasping, we can utilize position control to move the end-effector in the direction of $\tau^{3D}$, or leverage impedance control in the real world based on the real-time impedance force feedback as in [70] to adjust the end-effector and safely perform the actions.

## 4  Experiments

### 4.1  Experimental Setup

To verify the effectiveness of the proposed method, we conduct extensive evaluations in both simulation and real-world settings.

In simulation, we adopt IsaacGym [71] as the simulator, GAPartNet [5], and YCB [72] datasets as object assets, and a flying Franka Panda gripper for manipulation. We collected over 70 objects in 10 categories and evaluated them on 3 kinds of tasks, including Open, Close, and Pickup, comprising 13 different tasks. For each task, we conduct 50 experiments. The camera viewpoint is fixed, and the objects' positions and rotations are randomly initialized within a certain range. For Pickup tasks, several distractor objects are also randomly placed. A manipulation success is defined by whether the DoF of interest (articulation joint or height) exceeds a threshold.

In the real-world setting, we conduct experiments that involve interacting with various real-world household objects. We employ a Franka Emika robotic arm with a parallel gripper and utilize an on-hand RealSense D415 camera to capture the RGBD image. In addition, we also utilize a Unitree B1 robot dog equipped with a Z1 arm and a RealSense D415 camera to show our method's cross-embodiment nature [73, 74], which is covered in §4.5.

### 4.2  Baseline Methods

We compare our method against three baselines, namely **Where2Act** [18], **VRB** [16], and **Robo-ABC** [55], where they represent training-based, 2D affordance-based and retrieve-and-transfer methods respectively. Where2Act [18] and VRB [16] are trained on vast amounts of point clouds and egocentric HOI images, respectively, while Robo-ABC and our method are training-free.

Note that VRB and Robo-ABC only predict 2D affordance in either contact point and direction (VRB) or contact point only (Robo-ABC). To adapt them to the 3D space, we faithfully conduct necessary modifications. For Robo-ABC, we feed it with our collected affordance memory and use the proposed 2D affordance transfer module to generate a 2D trajectory. Subsequently, we follow the same procedure as in our method to lift 2D affordance to 3D for both methods. A detailed discussion of baseline implementation can be found in the appendix.

### 4.3  Results and Analysis

For simulation and real-world settings, we adopt Success Rate **(SR)** as the major evaluation metric. Results of simulation experiments are shown in Table 1, from which we can see that our method outperforms all baselines in the vast majority of tasks, yielding an average success rate of 52.62%. Compared to our method, Where2Act [18] largely fails in the opening tasks that require a precise contact point and grasp pose.

| Object | | | | | | | AVG |
|---|---|---|---|---|---|---|---|
| Task | O | O | O | P | P | P | / |
| Robo-ABC* [55] | 2/5 | 1/5 | 1/5 | 3/5 | 4/5 | 4/5 | 50.0 |
| **RAM (Ours)** | **3/5** | **2/5** | **3/5** | **3/5** | **4/5** | **5/5** | **66.7** |

Table 2: Success rates of object manipulation for different methods in the real-world environment. RAM yields a favorable real-world performance.

VRB [16] also suffers from providing precise contact points for grasping, leading to suboptimal performance in opening and picking up tasks. Our reimplemented and improved Robo-ABC [55] yields the second-best performance. We further note such a performance of Robo-ABC significantly depends on our affordance transfer and lifting modules.

We also compare our method with Robo-ABC [55] in the real world on 6 tasks with 5 rollouts each, as shown in Table. 2. Results show that our method also outperforms the baseline due to our more effective retrieval pipeline. It also shows our method's cross-domain nature and easy sim-to-real transfer. Real-world videos can be found in the appendix.

Apart from quantitative studies, we also show some qualitative results on affordance transfer with respect to multiple data sources in Fig. 5. Visualization results show that our method can establish a more reliable affordance transfer, leading to more robust performance.

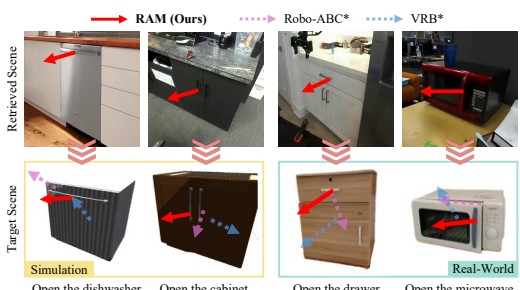

Figure 5: Qualitative comparison of generated 2D affordance. Source demonstrations retrieved by RAM are shown in the first row.

### 4.4 Ablation Studies

Probing deeper into our framework designs, we perform extensive ablation studies on various components. All studies are conducted on `Open` tasks of **drawers, cabinets, and microwaves**, with 20 episodes each. The metric of Distance to Mask **(DTM)** is reported along with SR as in [55], which measures the distance of the transferred contact point to the object handle in pixel space.

**Ablation on Hierarchical Retrieval.** As shown in Table 3, the removal of any submodules in the hierarchical retrieval pipeline results in a degradation of SR and DTM. We further observed the deactivation of geometrical retrieval leads to a significant SR drop from 38.3% to 26.7%, showing the importance of geometrical alignment in affordance transfer.

**Cross-Task Transfer.** Although our affordance memory construction pipeline enables scalable skill acquisition from various data sources, it is still valuable to probe how RAM performs when the target task is beyond the scope of its memory. We evaluate this by blocking the given task from the affordance memory. As shown in the 4th row of Table 3, RAM maintains a decent success rate of 25.0%, demonstrating an impressive cross-task transfer capability when facing unseen tasks.

| Ablation Method | SR ↑ | DTM ↓ |
|---|---|---|
| w/o Task Rtrvl. | 31.7 | 4.99 |
| w/o Sem. Filtering | 33.3 | 6.69 |
| w/o Geom. Rtrvl. | 26.7 | 9.36 |
| Cross-Task | 25.0 | 9.33 |
| SD → CLIP [56] | 15.0 | 57.4 |
| SD → DINOv2 [66] | 23.3 | 5.09 |
| SD → SD-DINOv2 [63] | 36.7 | **3.04** |
| **Full Pipeline** | **38.3** | 3.39 |

Table 3: Ablation studies on different components of our method.

**Effects of Different VFMs.** We further replaced Stable Diffusion (SD) used in our pipeline with other visual foundation models to investigate the effects of different VFMs for affordance transfer. As depicted in Table 3, we found that SD [57] performs best on SR, and SD-DINOv2 [63] performs best on DTM, both surpassing CLIP [56] and DINOv2 [66] by a large margin. Although SD-DINOv2 [63] excels in contact point matching, we choose SD for our pipeline due to its robustness in post-contact correspondence, higher success rate, and less inference time.

**Effects of Data Amount.** Ablation on the data amount in our retrieval memory $\mathcal{M}$ is conducted as shown in Fig. 6. We reduced the retrieval memory for each task from a minimum of 10% to full data. The results indicate that as the data amount increases, the overall performance also improves. Notably, using 50% of original data yields comparable results to using full data, demonstrating the data efficiency of RAM.

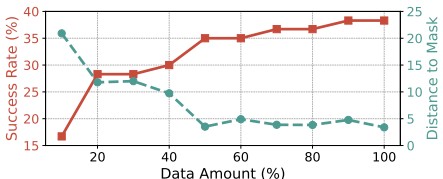

Figure 6: Performances of different data amounts for retrieval.

### 4.5 Downstream Applications and Discussions

In this section, we aim to show that our method has the potential to enable a broad spectrum of downstream applications, tackling general robotic problems.

**Policy Distillation.** As a zero-shot robotic manipulation method, conditioned on language instructions, our RAM can perform fully autonomous exploration of the surrounding environment by effective trial-and-error without any human priors or reward shaping. Therefore, we can use the proposed RAM for efficient annotated data collection and learn an end-to-end policy from the affordance knowledge. To that end, we leverage our zero-shot pipeline to automatically collect successful

| Task | 🗄 | ⊞ | 🖵 | AVG |
|------|------|------|------|------|
| VRB [16] | 1/20 | 1/20 | 2/20 | 6.67 |
| Zero-Shot | 8/20 | 4/20 | 6/20 | 30.0 |
| Distilled | **12/20** | **14/20** | **13/20** | **65.0** |

Table 4: Policy distillation results on success rate. RAM performs fully autonomous exploration to collect high-quality demonstrations efficiently for policy learning.

demonstrations of opening a drawer, a cabinet, or a microwave. Then, we learn an ACT policy [39] from these demonstrations. As shown in Table 4, using only 50 demonstrations for each task, our distilled policy can achieve a +35.0% performance boost over the zero-shot pipeline, indicating the huge potential for automatic and scalable high-quality data collection for policy learning.

**One-Shot Visual Imitation with Human Preference.** Apart from utilizing out-of-domain demonstration retrieval for manipulation, our method is naturally adaptable for one-shot visual imitation for better controllability, given a specific in-domain or out-of-domain demonstration. For example, as shown in Fig. 1c, given a tissue box with human preferences of picking up the tissue *paper* or tissue *box*, our method can act correspondingly to perform different visual imitations. Another example of *Tom and Jerry* shows that our method is able to bridge the great domain gap between the real world and cartoon images, thanks to the generalizability of visual foundation models.

**LLM/VLM Integration.** Our method can also be easily integrated with LLMs/VLMs for open-set instructions [75] and long-horizon tasks, by decomposing them into smaller ones suitable for affordance transfer and other action primitives. As shown in Fig. 1d, given an instruction (*"Clear the table."*), we can leverage a VLM [76] to interpret and decompose it into several actions and primitives and leverage affordance transfer for certain actions to behave in a human-oriented way. This enables more flexible tasks with higher complexity. More details can be found in the appendix.

## 5 Conclusions and Limitations

**Conclusions.** We propose RAM, a novel pipeline for generalizable zero-shot robotic manipulation. At the core of RAM is a retrieval-based affordance transfer and lifting mechanism that effectively distills actionable knowledge from large out-of-domain data to an unseen target domain. This framework can potentially generalize to other robotic tasks beyond manipulation, highlighting its versatility. In addition to the major experiments where RAM outperforms its counterparts by large margins, we also demonstrate the flexibility of RAM through its applications in several key areas.

**Limitations.** Despite compelling results, RAM shares certain limitations with prior works. For long-horizon tasks, it takes multiple steps for our method to chain the actions, which might generate less natural long-horizon behaviors. Also, our method struggles with complex actions, such as screwing. Future works to address these issues include integrating better foundation models for task planning and developing more advanced affordance transfer methods that directly lift 2D trajectories into 3D space to enable more complex tasks.

## Acknowledgments

The authors express their sincere gratitude to Dieter Fox, Ruihai Wu, Yuanchen Ju, Jiazhao Zhang, Xiaomeng Fang for fruitful discussions and valuable feedback. This work is partly supported by a gift from Google.

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

# A  Real-Robot Rollouts

Dynamic videos of real-robot rollouts can be found in the supplementary video and at our website: https://yuxuank.com/RAM/.

# B  Data Collection and Affordance Extraction

## B.1  Robotic Data

We adopt DROID [2] as our source of robotic data, which includes 76,000 expert trajectories of robots conducting daily tasks, along with corresponding task instructions. To extract affordance information from DROID, we first query instructions for tasks of interest. For example, for the task "open the drawer," we perform a hard search to filter out instructions that do not include the action "open" and the object "drawer."

Next, the filtered instructions are sorted based on the L2 distances between their language embeddings and the query embedding. We generate these embeddings using the `text-embedding-3-small` model of OpenAI. Based on the sorted instructions, we then select the Top-$k$ episodes for further affordance extraction.

To extract affordance information from these selected episodes, we identify the cartesian position when the gripper is closed as the 3D contact point. We then track the gripper's position for the following 10 time steps or until the gripper stops moving for consecutive steps. This provides us with the 3D post-contact trajectory. Using the provided camera parameters, we project the 3D contact point and the post-contact trajectory onto the first frame of each episode where the object is unobscured. Note that some affordance demonstrations are further refined manually by adding an offset or being removed due to inaccurate camera parameters. Should accurate camera calibration be available, the manual correction is not necessary.

This method ensures a precise and reliable extraction of affordance information from the DROID [2] dataset. Note that this method can also be used to process other robotic datasets (real-world or synthetic), such as [1, 3, 4].

## B.2  HOI Data

In addition to the details in the main text, we further note that we only average the hand keypoints within the object mask to determine the contact point and shift the post-contact trajectory accordingly. This ensures that the contact point is within the object for robust affordance transfer.

Note that this affordance extraction procedure can also be applied to other data sources where hand-object interactions are involved, such as more HOI datasets [9, 10, 11], vast amounts of unannotated human egocentric videos on the Internet, and user-provided demonstrations.

## B.3  Custom Data

The annotation process for custom data affordance has been discussed in the main text. Additionally, the sources of custom data are highly diverse and configurable. For our experiments, we annotated objects of interest in images obtained from the Internet (Google, YouTube, etc.).

Notably, custom data can also come from a variety of sources, ranging from user-captured images, cartoon images, AI-generated content, and even sketches, etc., demonstrating the flexibility and diversity of our data sources. This flexibility allows for a wide range of potential applications and extends our affordance memory's scalability to a greater extent. Examples of our custom data are shown in Fig. 7, which are merely ordinary pictures of various objects.

## B.4  Affordance Memory Statistics

The statistics of our affordance memory can be found in Table 5.

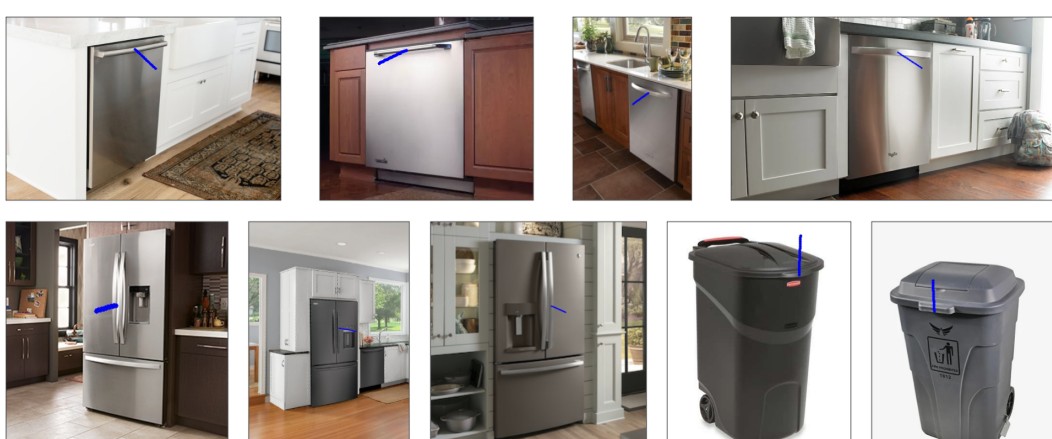

Figure 7: Examples of custom data affordance.

| Task Name | Icon | Data Source | Size |
|-----------|------|-------------|------|
| Open the drawer | ▤ | DROID | 30 |
| Close the drawer | ▤ | DROID | 20 |
| Open the cabinet | ▯ | DROID | 12 |
| Close the cabinet | ▯ | DROID | 6 |
| Open the microwave | ▣ | DROID | 42 |
| Close the microwave | ▣ | DROID | 10 |
| Open the dishwasher | ▤ | Custom | 10 |
| Open the refrigerator | ▯ | Custom | 20 |
| Open the trashcan | ▥ | Custom | 20 |
| Pickup the pot | ▭ | DROID | 11 |
| Pickup the mug | ▭ | HOI4D | 149 |
| Pickup the bowl | ▽ | HOI4D | 252 |
| Pickup the bottle | ▯ | HOI4D | 78 |
| Total | / | / | 660 |

Table 5: Affordance memory statistics.

## C   Implementation Details

### C.1   Feature Extraction Using Foundation Models

We use different foundation models as visual feature extractors, including:

- **Stable Diffusion (SD)** [57]. As illustrated in [62], given an original image $x_0$, we first add noise of time step $t$ to it to move it to distribution $x_t$, and then feed it to the stable diffusion network $f_\theta$ along with $t$ for denoising to extract the intermediate layer activations as the diffusion features (DIFT). We use the same configuration as in [62].
- **DINOv2** [66]. Extracting DINOv2 features is implemented by feeding the original image to the DINOv2 model and extracting the intermediate layer activations of DINOv2 ViT during the feed-forward process.
- **SD-DINOv2** [63]. As in [63], we first extract SD features and DINOv2 features and then do L2 normalization on them to align their scales and distributions. After that, we concatenate these two features together to get the SD-DINOv2 feature.
- **CLIP** [56]. Similar to DINOv2, We extract dense CLIP features by utilizing the intermediate layer activations of CLIP ViT.

## C.2 IMD Metric Calculation

As in [67], Instance Matching Distance (IMD) is originally proposed to examine pose prediction accuracy. Given a source image $I^\mathcal{S}$ and a target image $I^\mathcal{T}$, their normalized and masked feature maps $F^\mathcal{S}$ and $F^\mathcal{T}$, and a source instance mask $M^\mathcal{S}$, the IMD metric is defined as:

$$\text{IMD}(I^\mathcal{S}, I^\mathcal{T}, M^\mathcal{S}) = \sum_{p \in M^\mathcal{S}} \left\| F^\mathcal{S}(p) - \text{NN}(F^\mathcal{S}(p), F^\mathcal{T}) \right\|_2, \tag{3}$$

where $p$ denotes a pixel within the source instance mask, $F^\mathcal{S}(p)$ is the source feature vector at pixel $p$, and $\text{NN}(F^\mathcal{S}(p), F^\mathcal{T})$ denotes the nearest neighbor vector in the target feature map $F^\mathcal{T}$ with respect to the source feature vector. IMD measures the similarity of two images via the average feature distance of corresponding pixels [67]. Using IMD in the geometrical retrieval stage, we can accurately retrieve the demonstration where the object is oriented in the most similar way as in the observation.

## C.3 Baseline Methods

- **Where2Act** [18] is designed for articulated object manipulation only, which takes an object point cloud as input and predicts point-wise actionability scores, action proposals, and action scores with three separate models. Another drawback of this method is that it processes the point cloud in a task-agnostic way, leading to ambiguity of the generated affordance. We adopt it to the evaluation tasks by 1) randomly sampling the contact point from the predicted top-5 actionable points, 2) proposing 100 actions using the action proposal model, and 3) selecting the action with the highest action score.
- **VRB** [16] predicts the contact point and direction only on 2D images. To make it applicable in real manipulation tasks, we lift the estimated 2D affordance to 3D using our proposed sampling-based affordance lifting module.
- **Robo-ABC** [55] is initially designed for object grasping only, where only the contact point of a source demonstration retrieved by CLIP [56] feature similarity is transferred on the 2D image, followed by AnyGrasp [21] for grasp pose selection. For a fair comparison, we feed it with our collected affordance memory. To extend it for articulated objects, we use the proposed 2D affordance transfer module to transfer both the contact point and post direction. Subsequently, we follow the same procedure as in our method to lift 2D affordance to 3D.

# D Experiment Details

## D.1 Experimental Setup Details

In the simulation, we utilize a flying Franka Panda gripper for simplicity. We utilize cuRobo [23] motion planner for position control of the gripper.

In the real world, we adopt two different robotic systems. In the Franka Emika robotic arm setting, we leverage an on-hand RealSense D415 camera for RGBD perception and utilize *MoveIt!* [22] motion planner for the transformation from the target end-effector pose to joint position trajectories. In the Unitree robot dog setting, we leverage a Unitree B1 dog with a Z1 arm, along with a Robotiq 2F-85 parallel gripper. The RealSense D415 camera is also on-hand mounted, and we control the arm using the Z1 SDK for delta cartesian-space control.

For grasp generation, we utilize AnyGrasp [21] to produce grasp proposals, along with GSNet [20] with a relatively low graspness score threshold and collision threshold for more dense grasp proposals in case there is no grasp pose close enough.

### D.2 Additional Experiments

In this work, we primarily focus on object-centric manipulation and the transfer of primitive skills in a zero-shot manner. However, we also recognize the importance of evaluating our approach in more complex, cluttered environments.

To evaluate our method on more in-the-wild scenarios, we have conducted additional experiments in cluttered environments, such as the scene shown in Fig 8. As shown in Table 6, our method still outperforms VRB [16] and Robo-ABC [55], with performance dropping by only 6.7% compared to cleaner environments, demonstrating robustness and applicability in complex scenes.

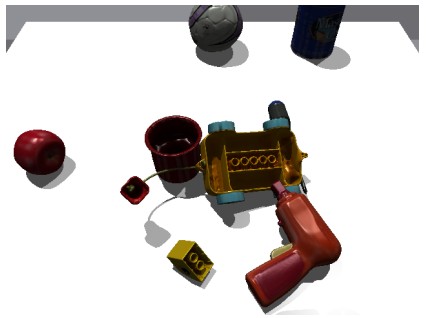

Figure 8: An example of a more cluttered table-top scene.

| Object | 🍵 | 🥣 | 🧴 | AVG |
|---|---|---|---|---|
| VRB* [16] | 26 | 30 | 50 | 35.3 |
| Robo-ABC* [55] | 40 | 40 | 52 | 44.0 |
| **Ours** | **50** | **66** | **56** | **57.3** |
| Ours (Clean) | 56 | 72 | 64 | 64.0 |

Table 6: Experiment results on more cluttered scenes.

## E  Failure Analysis

As a holistic robotic system that enables zero-shot manipulation in novel scenes, failure can happen in various stages of our pipeline. In our experiments, most failure cases stem from poor retrieval (e.g., extreme viewpoints or rare poses) or from inaccuracies in per-point transfer using diffusion features (e.g., slight deviations in the nearest neighbor points in the target image). We did not observe any specific object geometries consistently leading to failures. Additionally, our system encounters some common failure modes found in previous works, such as errors in grasp generation and motion planning.

To provide further context, we present visualizations of failure cases in Fig. 9. In the first scenario ("close the microwave"), the door of the target microwave is open at such an extreme angle that it becomes almost parallel to the line of sight. From this perspective, the door appears as a thin line, making it difficult to detect and resulting in suboptimal retrieval. In the second scenario ("open the cabinet"), the small, dark handles on the cabinet doors blend into the wood in low-light conditions, reducing their visibility. Consequently, the transferred contact point slightly misses the handle's exact position.

## F  Downstream Application Details

### F.1  Training ACT Policy

For policy distillation, we utilize an ACT policy [39] to perform imitation learning from our self-collected demonstrations. ACT is based on CVAE Transformer architecture and adopts the idea of action chunking to mitigate compounding errors that are common in behavior cloning (BC). More details can be found in their original paper [39].

We use 5 RGB views ($5 \times 640 \times 480 \times 3$) and the robot's proprioception as observation. We set the chunk size to 60, and the latent space dimension to 512. We use L1 loss plus KL divergence regularization for supervision. The number of training iterations is set to 200K, and we set the learning rate to $1 \times 10^{-5}$ and batch size to 8.

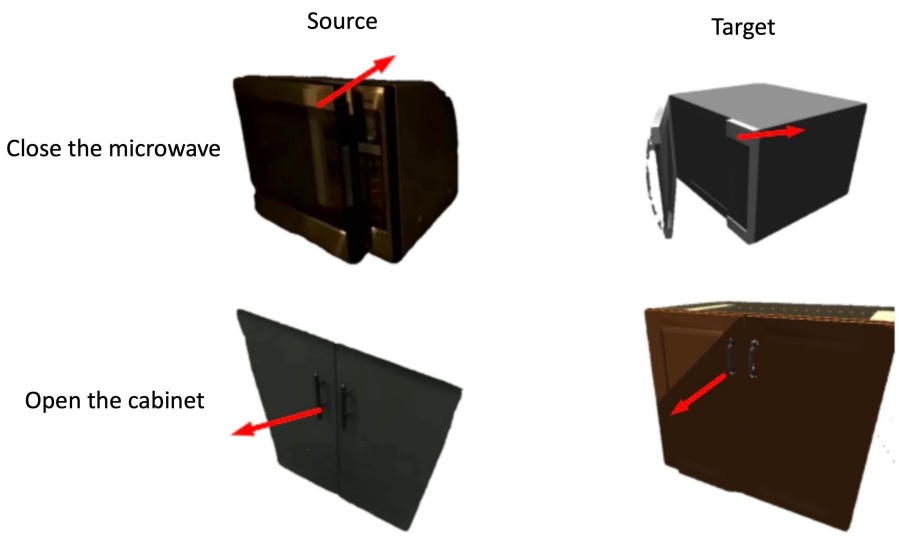

Figure 9: Visualization of failure cases.

## F.2 One-Shot Visual Imitation Details

For one-shot visual imitation conditioned on human preference, we pick out demonstrations either from our own in-domain demonstrations or from out-of-domain cartoon images (*Tom and Jerry* in this case). We ground and choose the first frame of interaction for $I^S$ and use the custom data annotation method to extract affordance manually. We then skip the hierarchical retrieval step and directly use these chosen demonstrations for affordance transfer and lifting, followed by 3D affordance execution.

## F.3 LLM/VLM Integration Details

For LLM/VLM integration, we utilize GPT-4V (`gpt-4-vision-preview`) [76] for task decomposition and scene understanding. We also use Grounded-SAM [77] for object detection and segmentation to produce 3D bounding boxes of objects in the scene.

Specifically, we define 3 basic primitives: `grasp()`, `move_to()`, and `release()` for VLM output. Note that these three primitives do not involve heuristics on specific object manipulation. Other than these primitives, when the VLM finds out there are relevant demonstrations in the affordance memory, it will schedule the proposed RAM system as a retrieval-augmented module to perform the action as a whole, followed by other defined primitives.

An example of our prompt and the VLM output is shown in Fig. 10.

```
===========
You are an intelligent robot dog that has an arm with a parallel gripper
for object manipulation.

You are given a human instruction and a scene observation. Your task is
to correctly manipulate the objects safely conditioned on the
instruction.

===========
```

You have a series of primitives and demonstrations you can leverage to perform the task. Based on the instruction, you can freely decompose it into several sub-tasks that are easier to finish and then chain them together.

First, you are endowed with 3 primitives, which are:
1. grasp(), which takes in object name, detects the object, moves to a graspable pose, and closes the gripper.
2. move_to(), which takes in a 6D pose and does motion planning to it.
3. release(), open the gripper to release the holding object.

The calling format should be like release(), etc.

Apart from the primitives, you also have an affordance memory, which contains lots of tasks that already have plenty of demonstrations. You are also equipped with a powerful robotic system called RAM, which can utilize those demonstrations in your affordance memory to directly perform corresponding sub-tasks, without calling the primitives. The tasks in your affordance memory are:

- Open: drawer, cabinet, microwave, dishwasher, refrigerator, trashcan
- Close: drawer, cabinet, microwave
- Pickup: pot, mug, bowl, bottle

The calling format should be like Open_drawer(), etc.

For the instruction and observation, please output a sequence of actions or primitives in the format of function calling as above. And finally, you should convert the sequence into a python list for convenience.

```
===========

A basic example:

Human instruction: Please open the drawer and pick up the utensil.
Observation: A closed drawer.

Your output:
<agent>
+++
First, I will decompose the instruction into several sub-tasks. The instruction is 'Please open the drawer and pick up the utensil.', so I need to
first open the drawer, and then pick up the utensil.
Sub-tasks:
1. open the drawer
2. pick up the untensil
+++
Then, for each subtask, I will see whether it can directly be performed using my affordance memory.
For 1, there are demonstrations regarding opening drawers, so I will directly use it.
For 2, there are no demonstrations for utensils, but picking up bottle is similar. Although usable, I will use primitives to ensure safety.
Action sequence:
1. Open_drawer()
2. grasp(utensil)
3. move_to(utensil_above_position)
4. move_to(release_position)
5. release()
+++
Finally, I will convert the sequence into a python list:
<output>
['Open_drawer()', 'grasp(utensil)', 'move_to(utensil_above_position)', 'move_to(release_position)', 'release()']
</output>
+++
</agent>

===========

Now you need to clear this table, please find a feasible way of doing that. Follow the example above:
```

```
Human instruction: Please clear the table.
Observation: There is a bottle on the table, and a trashcan with cups and bottles next to it.

<agent>
+++
First, I will decompose the instruction into several sub-tasks. The instruction is 'Please clear the table', so I need to first pick up the bottle
from the table, move it to the trashcan, and release it.
Sub-tasks:
1.Pick up the bottle from the table.
2.Move the bottle to the trashcan.
3.Release the bottle into the trashcan.
+++
Then, for each subtask, I will see whether it can directly be performed using my affordance memory.
For 1, there are demonstrations regarding picking up a bottle, so I will directly use it.
For 2 and 3, I will use the primitives to ensure the correct and safe movement to the trashcan and releasing the bottle.
Action sequence:
1.Pickup_bottle()
2.move_to(trashcan_position)
3.release()
+++
Finally, I will convert the sequence into a python list:
<output>
['Pickup_bottle()', 'move_to(trashcan_position)', 'release()']
</output>
+++
</agent>
```

Figure 10: Example prompt and VLM outputs of our LLM/VLM integration system.

