# OpenReview forum: "RAM: Retrieval-Based Affordance Transfer for Generalizable Zero-Shot Robotic Manipulation"
_robot-learning.org/CoRL/2024/Conference — CoRL 2024_

### Official Review · Reviewer_2wfH · 2024-07-07
**Great idea and presentation, few questions**

**Originality:** 4
**Technical Quality:** 3
**Clarity Of Presentation:** 5
**Potential Impact:** 3
**Recommendation:** 3
**Confidence:** 5

**Review:**

Strengths:

* The paper is very well-written and easy to follow, a pleasant read from beginning to end.

* The problem of generalizing to unseen tasks, environments and embodiments is indeed very important. The memory-augmented solution introduced in this paper is intuitive and well-designed. Another novel aspect is that it's training-free, of course only possible due to foundation models.

* The ablations are well-designed and interesting and the qualitative results (e.g. Tom and Jerry to robot) are impressive.

Weaknesses:

* One question in the method concerns the assumptions behind the coverage of the memory bank. Specifically, given a new scene and viewpoint, is a similar viewpoint for a similar scene expected to be in the memory bank? If no, how robust is the overall system to novel viewpoints?

* Another question is with regard to the retrieval capability. It seems that the different stages of the hierarchical retrieval are important. How robust is the retrieval pipeline when the scenes are very cluttered and contain a lot of objects? The current ablations examine objects in isolation, if I'm not mistaken, so this is not well-addressed. I assume this is why omitting the task retrieval does not lead to complete collapse, because in the examined dataset it's fair to assume the task based on the scene alone. In general, I'm wondering whether the proposed approach could generalize on a dataset where it's not obvious what is the next task to perform, such as CALVIN.

* Also related to the above, it is unclear how often the method retrieves, plans and executes. If the approach is not open-loop, then aren't the intermediate states OOD, i.e. not in the memory bank? I would expect temporal alignment to be an issue, unless the memory bank contains intermediate frames from the demonstrations as well. If that's the case, then I would expect the memory bank to grow very large. Some statistics on how large is the bank and how many frames on average are filtered at every retrieval stage would be interesting.

* The compared baselines are relatively weak. I'm wondering if more recent baselines could be included from the 3D manipulation literature, such as RVT (Goyal et al., 2023) or 3D Diffuser Actor (Ke et al., 2024). I understand that these won't be training-free and won't work on unseen tasks, but it's good to estimate where the current supervised SOTA would be to understand how well the proposed zero-shot approach works.

* Lastly, another useful baseline would be a model that is trained on a variety of tasks, environments and embodiments, to compare its capability to generalize to the attributes this submission aims. Specifically, Octo could serve as such a baseline.

**Quality Of The Limitations Section:**

3

**Questions For Rebuttal:**

See weakness above.

**Robotics Focus:**

4

**Summary Of Paper:**

The paper aims to build a robot agent that generalizes training-free to unseen tasks, environments and embodiments, using foundation models. The main argument is that collecting high-quality robot data for every task, environment and embodiment is infeasible. To this end, it proposes the collection of a bank of demonstrations from both robot videos as well as HOI videos. The bank contains image frames, paired with task descriptions and contact point trajectories. Given a new instruction, a relevant such triplet is retrieved from the bank and its point trajectories are mapped through correspondences in the current scene. The waypoints are then converted to actions. The method exhibits good results on articulated objects in sim and real, as well as several interesting ablations and applications.

**Summary Of Recommendation:**

I believe the paper is in a good state and has more merits than drawbacks. If the authors answer to the reviewers' questions it could be an even stronger submission.

---

### Official Review · Reviewer_38ey · 2024-07-21

**Originality:** 4
**Technical Quality:** 4
**Clarity Of Presentation:** 4
**Potential Impact:** 3
**Recommendation:** 3
**Confidence:** 5

**Review:**

The paper is well written, with a clearly explained method, an interesting overall approach and a well structured experiment section. The ability to retrieve and transfer a set of affordances and relative "short actions", and then transfer them to new situations, is demonstrated to be effective. Moreover, the paper compares RAM with a set of recent and well suited benchmarks, demonstrating the benefits brought by it. While the overall architecture is not entirely novel, as also noted in the paper, the additions and improvements over the existing literature justify its use.

Strengths:
- The paper is well written and motivated. All sections are well designed, from the method to the experiments.
- The method is effective, and the paper also investigates a series of design choices, like what Vision Foundation Model works best, therefore giving interesting insights to the reader.
- The experimental section justifies the design choices, and also compares RAM with a series of similar and therefore well suited methods from the recent literature.
- The ability to extract robot skills from not only robotics data is certainly remarkable and can help leveraging data of multiple forms.

Weaknesses:
- Related to the last point, it is not clear how effective the method would be with unlabelled HOI data, as it appears the method currently needs to use the labels from the HOI4D dataset (line 146).
- On a similar note, while it is interesting to see a skill extracted from a cartoon clip, the paper then specifies that the actual movement is manually annotated: therefore, the method simply performs a form of pose estimation of the cartoon drawer. This should be more clear in the website, that otherwise suggests that the method is able to infer everything simply from that short clip.
- While the method is effective in the proposed experiments, all the skills are relatively simple and composed of linear movements: picking up, pulling, pushing.

Minor points:
- It is not clear, in line 175, what segmentation masks the sentence refers to.
- I would double check the use of acronyms. In line 190 SD is used without being defined before.

**Quality Of The Limitations Section:**

3

**Questions For Rebuttal:**

1) How would you scale the method to trajectories more complex than the ones proposed?
2) While matching pre-contact/contact/post-contact points in 2D, some of these may end up in the background of the scene. When using VFM features to match it on a new image, how do you avoid this sticking to a random background position?

**Robotics Focus:**

4

**Summary Of Paper:**

The paper proposes RAM, a method to retrieve, from a dataset of robot data, human videos, and more, affordances and actions that can be applied to new scenarios.

**Summary Of Recommendation:**

I overall believe this is a good paper, which proposes both an interesting method, demonstrating the benefits with respect to similar techniques from the literature, and that also provides a series of interesting insights to the reader regarding how to effectively perform visual retrieval, specifically by leveraging Vision Foundation Models.

---

### Official Review · Reviewer_Xxnf · 2024-07-21
**Transferring action knowledge from diverse sources of data for robot-object interaction.**

**Originality:** 2
**Technical Quality:** 3
**Clarity Of Presentation:** 4
**Potential Impact:** 4
**Recommendation:** 4
**Confidence:** 4

**Review:**

Overall, the work is well-written, clearly expressed, and has thorough experimental results supporting the claim. I want to request that the authors highlight the limitations of the work better and provide a failure analysis. For example, adding failure analysis of retrieval (task/semantic/geometric) and mapping from 2D to 3D.

Strengths:
- While the idea of using affordance as a unified representation is not novel, the paper demonstrates its effectiveness using a simple retrieval-based approach.
- Well motivated, and ideas are clearly expressed
- Experimental setup and results reflect the method's effectiveness with thorough comparisons with baselines and ablations.

Weakness:
- Requires activity-segmented dataset with language annotation
- Retrieval can become the bottleneck as the size of the dataset increases as compared to learning to predict affordances.
- Evaluations are mainly performed on clean scenarios with one/two objects in the scene.
- Missing failure analysis.
- Limited technical novelty.

**Quality Of The Limitations Section:**

1

**Questions For Rebuttal:**

Why not fine-tune the vision models used for retrieval for affordance prediction instead of retrieval?

All the evaluation scenarios have one or two objects in the scene. Does the method also work in “in-the-wild” scenarios?

I wonder what the contribution of each individual dataset (robotic, HOI, custom data) is to the evaluations.

How would the method work with increasing dataset sizes? Wouldn’t retrieval become a bottleneck compared to learning to predict affordance?

While pre-processing human datasets, how does the method handle the occlusion of contact points?

Would the method's reproject from 2D to 3D work for articulated objects like sliding doors?

**Robotics Focus:**

4

**Summary Of Paper:**

The works proposes a unified representation for transfering action knowledge from robotic, human data, low-cost custom data.  The work proposes to us object affordance to transfer the knowledge of where and how to act. To enable the transfer of knowledge, it uses retrieval based method from a pre-processed data from a mix of robotic, human, custom data.

**Summary Of Recommendation:**

I recommend weak accept based on the concerns expressed in questions. Edit: I have updated my score to strong accept.

---

### Author Rebuttal · Authors · 2024-08-07

Dear AC and reviewers,

Thank you for your valuable feedback and recognition of our work’s strengths! We appreciate the reviewers’ acknowledgment of our well-written and clearly expressed paper, thorough experimental results, and innovative aspects of our memory-augmented, zero-shot solution for generalizable object manipulation under unseen objects, environments, and embodiments.

We highlight that RAM introduces a novel retrieve-and-transfer paradigm that leverages out-of-domain data to reduce reliance on costly in-domain demonstrations. Our hierarchical retrieval pipeline enhances accuracy by systematically retrieving relevant demonstrations from a diverse affordance memory. Additionally, the affordance lifting module converts 2D affordances into 3D actionable representations, ensuring compatibility with various robotic systems. RAM's embodiment-agnostic and data-efficient nature allows it to work across robotic platforms on the fly, demonstrating robust performance in novel scenes. These innovations distinguish our work and underscore its technical novelty.

The main issues raised by reviewers are: (1) the need for better highlighting of limitations and failure analysis, (2) potential bottlenecks in retrieval with large datasets, (3) the method’s effectiveness in cluttered and “in-the-wild” scenarios, (4) handling of unlabelled HOI data and complex trajectories, (5) robustness to novel viewpoints and cluttered scenes, and (6) comparisons with more recent baselines and generalist models like Octo.

We’ve addressed all concerns above and other minor points accordingly in the official comments. We hope our clarification can help to address your concerns.

**In the attached rebuttal file (in zip), we include:**

1) “cluttered_scene.png” that shows an example of more cluttered environments for additional experiments.

2) “sliding_door.png” that shows our method’s capability of manipulating sliding doors.

3) “sam2_HOI.mp4” that shows the newly released SAM-2 segmenting the HOI information from in-the-wild videos.

4) “sam2_cartoon.mp4” that shows the newly released SAM-2 segmenting the HOI information from Tom-and-Jerry cartoon videos.

5) "failure_visualization.png" that shows the visualization of failure case study.

6) "custom_data_examples.png" that shows some examples of the custom data.

Please refer to the official comments and the rebuttal file for more details. We sincerely hope our rebuttal can address your concerns. Thank you!

Best,

Authors

---

### Decision · Program_Chairs · 2024-09-04

**Decision:**

Accept

**Comment:**

**Before Rebuttal**

Strengths. The paper is well written, well structured, and easy to follow. The research and proposed method are well motivated, with a clear novel technical contribution. The ability to extract affordances from multiple modes of data (both human interactions and robot interactions) is helpful for scalability. There is a broad range of experiments, showing the strong performance of the proposed method with respect to sensible baselines, and exploring the effect of design choices. There are real-world experiments studying performance on realistic, everyday objects.

Weaknesses. The method requires the dataset to be manually segmented, which potentially limits the scalability. The skills the robot is able to perform have relatively simple trajectories (although this is acknowledged in the paper's limitations section). It is unclear how strong the generalisation is to novel viewpoints not contained in the dataset. There is only a limited amount of clutter and distractor objects in the scenes, so although experiments are done on realistic objects, it is not clear how well the method would work in more cluttered, everyday scenes. For real-world experiments, only 5 test episodes per task is quite low for drawing meaningful conclusions with statistical significance. The paper does not compare to training-free methods, so we are missing an understanding of how retrieval performs relative to explicit policy learning. The paper lacks an analysis of failure modes. The discussion of the limitations could be deeper; introducing the limitations by stating "RAM shares certain limitations with prior works" seems to avoid an honest exposition of limitations unique to RAM.

---

**After Rebuttal**

After the reviews, authors responded to the questions and concerns raised by the reviewers, and it was good to see some back-and-forth between authors and reviewers. After the rebuttal, the AC and reviewers had a discussion, where there was unanimous agreement the the paper should be accepted. Reviewers still have some minor concerns regarding the assumptions required for the method to work well, but overall this is an interesting paper with some technical novelty, and good experiments and ablations.